# Analysis of Differentially Expressed MicroRNAs in Serum and Lung Tissues from Individuals with Severe Asthma Treated with Oral Glucocorticoids

**DOI:** 10.3390/ijms24021611

**Published:** 2023-01-13

**Authors:** Marta Gil-Martínez, Clara Lorente-Sorolla, José M. Rodrigo-Muñoz, Miguel Ángel Lendínez, Gonzalo Núñez-Moreno, Lorena de la Fuente, Pablo Mínguez, Ignacio Mahíllo-Fernández, Joaquín Sastre, Marcela Valverde-Monge, Santiago Quirce, María L. Caballero, Francisco J. González-Barcala, Ebymar Arismendi, Irina Bobolea, Antonio Valero, Xavier Muñoz, María Jesús Cruz, Carlos Martínez-Rivera, Vicente Plaza, José M. Olaguibel, Victoria del Pozo

**Affiliations:** 1Immunoallergy Laboratory, Immunology Department, Instituto de Investigación Sanitaria Fundación Jiménez Díaz (IIS-FJD, UAM), Av. Reyes Católicos 2, 28040 Madrid, Spain; 2CIBER de Enfermedades Respiratorias (CIBERES), Instituto de Salud Carlos III (ISCIII), 28029 Madrid, Spain; 3Department of Genetics, Instituto de Investigación Sanitaria Fundación Jiménez Díaz (IIS-FJD, UAM), 28040 Madrid, Spain; 4Center for Biomedical Network Research on Rare Diseases (CIBERER), Instituto de Salud Carlos III (ISCIII), 28029 Madrid, Spain; 5Bioinformatics Unit, Instituto de Investigación Sanitaria Fundación Jiménez Díaz (IIS-FJD, UAM), 28040 Madrid, Spain; 6Biostatistics and Epidemiology Unit, Instituto de Investigación Sanitaria Fundación Jiménez Díaz (IIS-FJD), 28040 Madrid, Spain; 7Allergy Department, Hospital Universitario Fundación Jiménez Díaz, 28040 Madrid, Spain; 8Department of Allergy, Hospital Universitario La Paz, IdiPAZ, 28046 Madrid, Spain; 9Pulmonology Department, Complejo Hospitalario Universitario de Santiago, 15706 Santiago de Compostela, Spain; 10Allergy Unit & Severe Asthma Unit, Pulmonology and Allergy Department, Hospital Clínic, 08036 Barcelona, Spain; 11Pulmonology Department, Hospital Vall d’Hebron, 08035 Barcelona, Spain; 12Pulmonology Department, Hospital Germans Trias i Pujol, 08916 Badalona, Spain; 13Respiratory Medicine Department, Hospital de la Santa Creu i Sant Pau, 08041 Barcelona, Spain; 14Severe Asthma Unit, Department of Allergy, Complejo Hospitalario de Navarra, 31008 Pamplona, Spain; 15Department of Medicine, Faculty of Medicine, Universidad Autónoma de Madrid, 28029 Madrid, Spain

**Keywords:** biomarker, oral corticosteroids, miRNAs, individuals with severe asthma

## Abstract

Nowadays, microRNAs (miRNAs) are increasingly used as biomarkers due to their potential contribution to the diagnosis and targeted treatment of a range of diseases. The aim of the study was to analyze the miRNA expression profiles in serum and lung tissue from patients with severe asthma treated with oral corticosteroids (OCS) and those without OCS treatment. For this purpose, serum and lung tissue miRNAs of OCS and non-OCS asthmatic individuals were evaluated by miRNAs-Seq, and subsequently miRNA validation was performed using RT-qPCR. Additionally, pathway enrichment analysis of deregulated miRNAs was conducted. We observed altered expression by the next-generation sequencing (NGS) of 11 miRNAs in serum, of which five (hsa-miR-148b-3p, hsa-miR-221-5p, hsa-miR-618, hsa-miR-941, and hsa-miR-769-5p) were validated by RT-qPCR, and three miRNAs in lung tissue (hsa-miR-144-3p, hsa-miR-144-5p, and hsa-miR-451a). The best multivariate logistic regression model to differentiate individuals with severe asthma, treated and untreated with OCS, was to combine the serum miRNAs hsa-miR-221-5p and hsa-miR-769-5p. Expression of hsa-miR-148b-3p and hsa-miR-221-5p correlated with FEV_1_/FVC (%) and these altered miRNAs act in key signaling pathways for asthma disease and the regulated expression of some genes (*FOXO3*, *PTEN*, and *MAPK3*) involved in these pathways. In conclusion, there are miRNA profiles differentially expressed in OCS-treated individuals with asthma and could be used as biomarkers of OCS treatment.

## 1. Introduction

Asthma is a heterogeneous disease of the lower airways that affects approximately 300 million people worldwide [1]. Chronic airway inflammation as well as airway damage and remodeling are hallmark features of the disease and induce bronchial hyper-reactivity and variable expiratory airflow limitation [2]. Individuals with asthma exhibit a broad spectrum of disease severity. Severe asthma, defined as that which requires treatment with high-dose inhaled corticosteroids plus a second controller and/or systemic corticosteroids [3], affects about 5–10% of the asthmatic population, and is associated with increased mortality and morbidity, diminished quality of life, and greater health expenditure [4].

The pathophysiology of asthma is highly complex, comprising distinct phenotypes and endotypes [5]. In addition, a broad range of triggers including environmental risk factors, genetic polymorphisms, and epigenetic changes contribute to the onset of asthma, variations in phenotypes, and response to steroid treatment [6]. Severe asthma comprises two predominant inflammatory endotypes: Type 2 (T2)-high and T2-low disease [7]. T2-high asthma is more severe and difficult to treat, and many patients with this endotype require high-dose treatments and/or biological therapy for better control of the symptoms and exacerbations, and to avoid adverse reactions caused by the administration of oral corticosteroids (OCS) [8,9].

To help guide the selection of treatment in patients with severe asthma, currently available biomarkers such as peripheral blood and induced sputum eosinophil count, fraction of exhaled nitric oxide (FeNO) level, and immunoglobulin-E (IgE) levels have been used [10]. However, interest in using microRNA (miRNA) profiles as biomarkers for diseases is currently on the rise [11]. These small, single-stranded, non-coding RNA molecules [12] participate in the regulation of gene expression by inhibiting protein translation [13] and are useful as biomarkers as they are stable and disease-related particles [14]. Previous studies have demonstrated that miRNAs hold great potential as asthma mediators and biomarkers and can further be of use in asthma endotyping and phenotyping, thereby making it possible to offer personalized therapies for patients [15,16]. Among them, Mirra et al. recently identified a new expression profile of circulating miRNAs including a member of the miR-181 family, related to lung inflammation, which could be used as a clinical marker of bronchial asthma and response to pharmacological treatment [17].

The aim of this study was to search for a profile of differentially expressed miRNAs between individuals with severe asthma receiving OCS treatment versus those without OCS therapy using next-generation sequencing (NGS). The miRNAs found could serve as lung-tissue or systemic biomarkers of oral corticosteroid treatment and facilitate the selection of the most appropriate management approach. 

## 2. Results

### 2.1. Characteristics of the Study Population

Descriptive data reflecting the demographic, inflammatory, functional, and clinical characteristics of all individuals studied are summarized in Table 1 and Table 2. The subjects were distributed into two different analyses: 20 OCS-treated versus (vs.) 26 non-OCS-treated severe asthmatics (serum samples analysis) and six OCS-treated vs. seven non-OCS-treated asthmatic subjects (lung biopsy samples analysis).

As shown in Table 1, regarding inflammatory parameters, the OCS-treated and non-OCS-treated severe asthmatics from which serum samples were used had significant differences in IgE levels (366.0 and 96.8 IU, *p* < 0.05). Additionally, as expected, we observed significant differences in treatment with systemic corticosteroids (*p* < 0.0001). Other characteristics (i.e., demographic, inflammatory, functional, clinical) did not present statistically significant differences.

Moreover, comparing the treated and non-treated with OCS groups in the analysis of lung biopsy samples (Table 2), the only significant differences were detected in the percentage of monocytes (10.0% vs. 5.9%, *p* < 0.05) and, as planned, in the oral corticosteroid treatment (*p* < 0.01).

### 2.2. MiRNA Profile in Serum Samples

Analysis of the data obtained by NGS (miRNAs-seq) showed 11 differentially expressed serum miRNAs (hsa-miR-148b-3p, hsa-miR-221-5p, hsa-miR-618, hsa-miR-200b-3p, hsa-miR-548I, hsa-miR-200a-3p, hsa-miR-941, hsa-miR-181d-5p, hsa-miR-769-5p, hsa-miR-133a-3p, and hsa-miR-3614-5p) between the OCS-treated and non-OCS-treated patients with severe asthma (adjusted *p* < 0.05) (Figure 1a). All of these 11 miRNAs were overexpressed in the OCS-treated subjects. 

When PCA was performed using the miRNA expression values, the results revealed that both groups were clearly differentiated (Figure 1b).

After the miRNAs-Seq analysis, RT-qPCR validation of miRNAs was performed in a larger cohort of serum samples. Of the 11 differentially expressed miRNAs obtained by miRNAs-Seq between the OCS-treated and non-OCS-treated individuals with severe asthma, we found significantly increased levels in five of them (hsa-miR-148b-3p, hsa-miR-221-5p, hsa-miR-618, hsa-miR-941, and hsa-miR-769-5p) in the OCS-treated patients (Figure 1c). Hsa-miR-3614-5p was not detected by RT-qPCR, and the remaining miRNAs evaluated did not reach statistical significance (Figure 1c).

These results confirm that serum hsa-miR-148b-3p, hsa-miR-221-5p, hsa-miR-618, hsa-miR-941, and hsa-miR-769-5p were differentially expressed miRNAs between the OCS-treated and non-OCS-treated subjects with severe asthma, suggesting that these could serve as markers of treatment.

### 2.3. MiRNAs Correlation and Signaling Pathways

In order to establish some relation between the expression levels of these serum miRNAs (ΔCt) and different clinical characteristics, we performed correlation analyses. We observed a significant direct correlation between FEV_1_/FVC and hsa-miR-148b-3p in the two groups (correlation coefficient (r) = 0.63 for the OCS-treated patients with severe asthma or correlation coefficient (r) = 0.59 for non-OCS-treated subjects with severe asthma) and hsa-miR-221-5p among the OCS-treated individuals with severe asthma (correlation coefficient (r) = 0.56) (Figure 2a). Since there is a correlation between FEV_1_/FVC and hsa-miR-221-5p in the OCS-treated severe asthmatics, it can explain the difference in treatment between the two groups, whereas, since FEV_1_/FVC correlated with hsa-miR-148b-3p in both groups, we can say that, in this case, there is no explanation for the difference in treatment, but there was a link with asthma pathology. Furthermore, despite not being a result of statistical significance, it is worth noting the opposite trend between the groups of the OCS-treated and non-OCS-treated asthmatic individuals, in the correlation of hsa-miR-148b-3p with the FeNO levels, perhaps because OCS treatment modified the FeNO/miR-148b-3p relation.

ROC curves were generated, and we calculated the areas under the curve (AUC). Multivariate logistic regression models (continuous predictors) were created for better discrimination of the OCS-treated and non-OCS-treated patients with severe asthma (Appendix A). Hsa-miR-221-5p and hsa-miR-769-5p showed AUC values of 0.75 and 0.72, respectively, indicating that both are acceptable as univariate predictors. Furthermore, when both serum miRNAs were combined, the AUC was even better (AUC = 0.77), as this was the best multivariate logistic regression model for differentiating OCS-treated and non-OCS-treated severe asthmatics (Figure 2b).

When an in silico analysis was carried out to determine their involvement in the biological processes with the differently expressed miRNAs, we determined that these miRNAs are involved in the regulation of several crucial asthma pathways, which are extracellular matrix (ECM)–receptor interaction, fatty acid biosynthesis, steroid biosynthesis, the Hippo signaling pathway, and adherens junction (Figure 3a). As *TGFRB1*, *FOXO3*, and *PTEN* are target genes of hsa-miR-148b-3p, and *MAPK3* is the objective gene of hsa-miR-221-5p, all are involved in the signaling pathways related to asthma disease. The RT-qPCR assay was conducted in lung tissue samples from asthmatic patients and healthy subjects, revealing that *FOXO3*, *PTEN*, and *MAPK3* were significantly increased in asthmatic individuals compared to the controls (Figure 3b). 

Furthermore, to verify the association between miRNAs and the expression levels of their respective target genes (ΔCt), correlation analyses were carried out. We found a significant positive correlation for the total population between hsa-miR-221-5p and their target gene, *MAPK3* (Spearman r = 0.35) (Figure 3c). 

Finally, we examined the relation between the miRNA expression levels and eosinophil counts. We showed a statistically significant direct correlation between hsa-miR-221-5p with the percentage of eosinophils in the different population subsets (i.e., total subjects and asthmatic individuals) (Figure 3d).

### 2.4. MiRNA Profile in Lung Samples

Therefore, validated serum miRNAs were analyzed in the target tissue of asthma disease. We found no statistically significant differences in the expression of these miRNAs in the lung biopsies. For this reason, we decided to conduct miRNA-Seq in these samples from asthmatic patients in order to understand the miRNA profile in lung tissue. Here, three miRNAs were differentially expressed between the OCS-treated and non–OCS-treated asthmatics (*q* ≥ 8). Hsa-miR-144-3p, hsa-miR-144-5p, and hsa-miR-451a were overexpressed in the OCS-treated patients (Figure 4). 

We then performed RT-qPCR on asthmatic lung biopsies from a larger cohort to validate the miRNAs obtained by sequencing analysis. None of the three miRNAs were validated, however, we observed a similar expression tendency in OCS-treated asthmatic patients (2^−ΔΔCt^ for each hsa-miR: 144-3p = 6.5, 144-5p = 5.2 and 451a = 2.0), as obtained in miRNA-Seq.

## 3. Discussion

This article draws on data from a comprehensive study of the miRNAs differently expressed by NGS in the serum (of which 5 were validated: hsa-miR-148b-3p, hsa-miR-221-5p, hsa-miR-618, hsa-miR-941, and hsa-miR-769-5p) and in lung tissue (hsa-miR-144-3p, hsa-miR-144-5p and hsa-miR-451a) between the OCS-treated and non-OCS-treated patients. These miRNAs could be used as epigenetic biomarkers to distinguish individuals with severe asthma under OCS treatment, although we do not know whether this miRNAs alteration is due to treatment or a characteristic of patients that need OCS in their asthma control.

Asthma is a heterogeneous disease comprising different phenotypes and endotypes, and as a result, patient stratification by inflammatory endotype is a central component of the algorithm used to assess and manage severe disease [18,19]. Identification and application of biomarkers used to identify phenotypes and endotypes of severe asthma and guide therapeutic strategy is of increasing interest to clinicians, as individual approaches to disease management and personalized medicine are imperative [20]. The severe asthma endotype is currently defined by biomarkers of varying accuracy including blood and sputum eosinophils, periostin, FeNO, and IgE for T2 asthma [16,21,22]. As reported in the literature, we found that serum samples of OCS-treated asthmatic patients had elevated IgE levels compared to the non-treated asthmatics [23]. Here, it is worth mentioning that the expression of biomarkers may be modified or even suppressed by pharmacological treatment (i.e., oral corticosteroids for blood eosinophils), making it advisable to examine biomarkers in combination. For example, we observed that the combination of two serum miRNAs used as biomarkers (hsa-miR-221-5p and hsa-miR-769-5p) resulted in a better AUC than when these were analyzed individually, suggesting that the profile of these two miRNAs was greater than using them separately.

Biomarkers are promising tools for the recognition and management of non-adherence to inhaled and oral corticosteroid therapy in asthma and can help to identify potential responders to therapeutic options, thus enabling their use as predictors of treatment response [24]. In recent years, miRNAs have been proposed as diagnostic and prognostic biomarkers for diseases [25]. NGS is a valuable technique in these efforts and can be employed for whole genome analysis including mRNA and small RNA expression such as miRNAs [26]. Our group previously used NGS to describe 15 miRNAs differentially expressed between eosinophilic and non-eosinophilic asthmatic patients, two of which were validated and could serve as instruments to classify patients into different phenotypes/endotypes [15]. In addition, hsa-miR-144-3p was previously described by our group to present an increase in miRNA levels in corticosteroid treated subjects by RT-qPCR [27], which validates the result obtained in miRNAs-Seq from the lung biopsy samples of asthmatics. 

In this study, we found different profiles of miRNAs in the serum and lung tissue. Dysregulation of hsa-miR-148b-3p, hsa-miR-221-5p, hsa-miR-618, hsa-miR-941, hsa-miR-769-5p, hsa-miR-144-3p, hsa-miR-144-5p, and hsa-miR-451a has been previously associated with different aspects of asthma pathogenesis and/or other respiratory diseases [28,29,30,31,32]. Nevertheless, these profiles have never been described as indicators of treatment. These miRNAs could be relevant molecules with implications for the phenotypic distinction of patients based on whether they are treated or untreated and may be indicated for use in asthma pathogenesis, since they were significantly correlated with clinical parameters. Interestingly, we found different miRNA profiles in asthmatics regarding the sample origin, which underscore the roles of these molecules as epigenetic regulators, as found in previous studies of miRNA tissue specificity [33]. 

Furthermore, we observed an enrichment in the ECM–receptor interaction pathway, which is widely associated with asthma due to the central role it plays in the development of airway inflammation and remodeling associated with asthma [2]. Moreover, miRNAs have been shown to affect the expression of multiple genes directly or indirectly, and for this purpose, we studied target genes related to differential miRNA expression. Here, *FOXO3*, *PTEN*, and *MAPK3* exhibited increased expression in asthmatic samples, which correlated with the miRNA expression studied, which emphasizes the implication of these inflammatory genes in the disease, as previously described by others [34,35,36]. In addition, in this study, we observed a positive correlation between hsa-miR-221-5p and *MAPK3*, its target gene, instead of detecting, as usual, an inverse miRNA-target gene correlation. Given that miRNAs generally behave as silencers of their target mRNAs, a possible explanation would be that miRNA–mRNA relationships are complex, and the lung tissue samples in which they have been studied differ from each other (influence of genetic and/or environmental factors), while there may be other miRNAs involved in the modulation of a particular mRNA (*MAPK3* mRNA), and there could also be interaction of other target genes of this miRNA (hsa-miR-221-5p) that could be inhibitors of *MAPK3*, in this case, remarking the intricacy of miRNA–mRNA post-transcriptional regulation in whole tissue.

One of the main limitations of this study is the lack of the same-subject pre-treatment samples, which made it impossible to determine whether the difference in the miRNA profiles between the groups of patients treated with OCS therapy was a consequence of the treatment (reflects treatment response) or if, in contrast, the existence of differentially expressed miRNAs between the groups was a baseline characteristic of the disease and selected the treatment. Another limitation is the low number of lung biopsy samples available from asthmatic patients and, in fact, some of these samples were derived from individuals with other severe lung pathology, which could probably explain the reason why no significant differences were observed. Due to the limited number of analyzed lung biopsy samples available from the asthmatic patients, it is a preliminary level study in this case. Finally, it would be very interesting to evaluate the levels of protein expression by ELISA of the genes regulated by two of the altered miRNAs (hsa-miR-148b-3p and hsa-miR-221-5p) in this study.

## 4. Materials and Methods

### 4.1. Study Subjects and Sample Collection

Serum samples were obtained from 46 subjects with a severe asthma diagnosis who were recruited from allergy and pulmonology units of a series of hospitals in Spain; from these, we selected samples from 20 patients undergoing treatment with OCS and 26 not receiving OCS. MiRNA-sequencing (miRNA-Seq) was performed on serum samples from six of the OCS-treated and seven non-OCS-treated patients, respectively. Of the remaining 33 samples, 14 of those obtained from OCS-treated and 19 from non-OCS-treated subjects were included for miRNA validation by means of quantitative real-time polymerase chain reaction (qPCR). Descriptive data representing the demographic, inflammatory, functional, and clinical characteristics of the study subjects were collected.

All patients were participants in the MEGA project, which studies a cohort of asthmatic individuals with varying grades of severity [37]. The inclusion criteria were as follows: (i) acceptance to participate by providing signed informed consent; (ii) asthma diagnosis following the 2021 GINA criteria [38]; and (iii) age between 18 and 75 years. The definition of OCS-treated and non-OCS-treated severe asthmatic was established based on their treatment.

The study was conducted in accordance with the tenets of the Declaration of Helsinki, and the protocol was approved by the participating hospital ethics committees.

In addition, we analyzed the lung biopsy samples and relevant descriptive data corresponding to 13 asthmatic patient donors; these were also subdivided depending on whether the individuals were treated with OCS (OCS-treated) or not (non-OCS-treated). Furthermore, we had 20 lung biopsy samples from the control subjects, required for the gene expression analysis in lung tissue. Lung biopsy samples from these asthmatic patients and control individuals were provided by the CIBERES Pulmonary Biobank Consortium (PT13/0010/0030), a network currently formed by 12 tertiary Spanish hospitals (www.ciberes.org, accessed on 10 January 2023) detailed in the Acknowledgements section, and integrated in the Spanish National Biobanks Network. Lung biopsies were processed in accordance with the standard operating procedures, and subsequent approval was granted for processing by the local ethics and scientific committees. 

Serum samples were obtained by blood clotting in anticoagulant-free tubes and subsequent centrifugation at 3000 rpm for 10 min at 4 °C and stored at −80 °C until use. Lung tissue samples were preserved in the RNAlater stabilization solution at −80 °C until use.

### 4.2. Library Preparation, MiRNAs-Seq, Bioinformatic Analysis

Serum RNA (including miRNAs) was extracted from 200 μL of serum using the miRNeasy Serum/Plasma Advanced Kit (Qiagen, Hilden, Germany), as indicated by the manufacturer. Lung tissue RNA was purified using the QIAzol Lysis Reagent (Qiagen, Hilden, Germany), followed by the application of the acid guanidinium thiocyanate-phenol-chloroform extraction method [39]. In all cases, the RNA enriched in miRNAs was eluted by adding 20 μL of RNase-free water.

Then, small RNA (miRNA-enriched RNA) was converted to Illumina sequencing libraries using the NEXTFLEX^®^ Small RNA-Seq Kit v3 (Bioo Scientific Corporation, Austin, TX, USA), strictly according to the manufacturer’s instructions. The size profile of the individual libraries was quantified using High Sensitivity D1000 Screentape on a 4200 TapeStation System (both Agilent, Santa Clara, CA, USA). Quantified libraries were sequenced on an Illumina MiniSeq 550 platform (Illumina, San Diego, CA, USA) using a MiniSeq 500/550 75-Cycle High Output Kit. 

Bioinformatic analyses including quality control, pre-processing, and statistical analysis of small RNA-Seq data were carried out by the Bioinformatics Unit of IIS-Fundación Jiménez Díaz. Adaptor removal and trimming of raw reads were performed using Cutadapt [40] by following the instructions for the NEXTflex small RNA-Seq Kit. Adaptor-trimmed reads between 17 and 25 nt were retained and aligned to the reference genome (GRCh38 assembly) using Bowtie2 [41]. Mapping of reads to known miRNAs was performed with HTSeq-count2 [42] using mature miRNA annotation retrieved from the miRBase database (miRBase v22). Raw miRNA counts across samples were normalized by sequencing depth and RNA composition using the trimmed mean of M-values (TMM) function from the NOISeq Bioconductor R package [43]. Subsequent principal component analysis (PCA) on normalized and scaled values was applied using the prcomp function from the R stats package [44]. Comparison of the normalized expression levels across groups was performed following two alternative methods for testing differential expression in the sequencing data: NOISeq [43] and DESeq2 [45]. Fold-change and *p*-values adjusted by false discovery rate (FDR) were calculated and used to identify significant differentially expressed miRNAs. MiRNAs were considered biologically relevant if they were differentially expressed (adjusted *p* < 0.05) and presented a Log2 fold change ≥1.5 between two groups, or if *q* ≥ 0.8 in the case of NOISeq.

### 4.3. RT-qPCR and Pathway Enrichment Analyses

For validation of the miRNA-Seq results or the expression analysis of the validated miRNAs, 4 µL of serum miRNA samples or 30 ng of lung tissue miRNA-enriched RNA were reverse transcribed to cDNA using the miRCURY LNA RT Kit (Qiagen), following the manufacturer’s protocol. The synthetic miRNAs SP6 and cel-miR-39-3p were used to control for correct reverse transcription. The cDNA obtained was stored at −20 °C until use. Then, miRNA expression was evaluated by qPCR using the miRCURY LNA SYBR Green PCR Kit (Qiagen), as indicated in the instructions. For this purpose, we used 3 μL of cDNA from the serum or lung tissue miRNAs diluted 1:60 or 1:30, respectively, in RNase-free water. Based on the results of the miRNA-Seq, the probes (Qiagen) used for the validation analysis of miRNAs in the serum and lung tissue were the following: hsa-miR-148b-3p, hsa-miR-221-5p, hsa-miR-618, hsa-miR-200b-3p, hsa-miR-548I, hsa-miR-200a-3p, hsa-miR-941, hsa-miR-181d-5p, hsa-miR-769-5p, hsa-miR-133a-3p, hsa-miR-451a, hsa-miR-144-3p, and hsa-miR-144-5p. Additionally, hsa-miR-103a-3p, hsa-miR-191-5p, SP6, cel-miR-39-3p, and U6 (Qiagen) were selected as housekeeping miRNAs. All samples were run in triplicate, and reactions were performed in a Light Cycler^®^ 96 thermocycler (Roche, Basel, Switzerland). Cycle threshold (Ct) values were analyzed with LightCycler^®^ 96 SW 1.1 (Roche) software and miRNA expression was calculated by normalizing to the endogenous miRNA controls by applying the 2^−ΔCt^ method [46], where ΔCt = Ct_miRNA_ − Ct_housekeeping miRNAs_. The relative quantification of differences in expression (RQ = 2^−ΔΔCt^; where ΔΔCt = ΔCt_OCS-treated_ − ΔCt_non-OCS-treated_) was carried out by the ΔΔCt method [46].

For gene expression analysis in lung tissue, 500 ng of RNA quantified by a Nanodrop ND-1000 spectrophotometer (Thermo Fisher Scientific, Waltham, MA, USA) was reverse transcribed using the High-Capacity cDNA Reverse Transcription Kit (Applied Biosystems, Foster City, CA, USA), followed by qPCR according to the manufacturer’s guidelines on a StepOne Real-Time PCR System (Applied Biosystems). TaqMan^TM^ gene expression probes were purchased for *TGFBR1*, *FOXO3*, *PTEN*, *MAPK3*, and *GAPDH* using TaqMan^TM^ Gene Expression MasterMix (Applied Biosystems). Gene expression was calculated by normalizing to the endogenous gene *GAPDH* control by applying the 2^−ΔCt^ method as previously reported (RQ = 2^−ΔΔCt^; where ΔΔCt = ΔCt_asthmatics_ − ΔCt_healthy subjects_) [46].

In order to identify target genes linked to asthmatic pathology of hsa-miR-148b-3p, hsa-miR-221-5p, hsa-miR-618, hsa-miR-941, and hsa-miR-769-5p, differentially expressed miRNAs in the serum between the OCS-treated and non-OCS-treated patients with severe asthma, pathway enrichment analysis of dysregulated miRNAs was performed using the DIANA-miRPath v3.0 bioinformatic online resource [47]. Relevant pathways for asthma disease were represented when a *p*-value < 0.05, and the related genes were analyzed by RT-PCR, as previously mentioned. 

### 4.4. Statistical Analysis

Statistical analyses and graphs were created with GraphPad Prism^®^ v6-8.0 (GraphPad Software Inc., San Diego, CA, USA) and R software^®^ v4.1.0 [44] (R Foundation for Statistical Computing, Vienna, Austria).

Results are shown as the mean (standard deviation, SD) or median (interquartile range, IQR) values. For all statistical analyses, differences showing *p* < 0.05 were considered significant. Normality was analyzed by means of the Shapiro–Wilk test. For continuous variables, comparisons of normally distributed data between non-paired groups were performed via an unpaired *t* test (equal SD) and *t* test with Welch’s correction (different SD), and non-normally distributed data and non-paired groups were compared by the Mann–Whitney test.

Additionally, correlations between the miRNA expression levels (ΔCt) and some clinical parameters (quantitative variables) were estimated by Spearman (non-normally distributed data) or Pearson (normally distributed data) correlation; the Fisher exact test was applied to a 2 × 2 contingency table to test the null hypothesis of the independence of groups and some clinical characteristics (qualitative variables). Finally, the expression profile (ΔCt) of each differentially expressed miRNA was used to create receiver operator characteristic (ROC) curves, and logistic regression models were developed to evaluate the performance of miRNA as biomarkers; an area under the curve (AUC) of 0.7 indicated an acceptable biomarker.

## 5. Conclusions

In summary, we describe the significant differences in the expression of eight miRNAs, hsa-miR-148b-3p, hsa-miR-221-5p, hsa-miR-618, hsa-miR-941, hsa-miR-769-5p, hsa-miR-144-3p, hsa-miR-144-5p, and hsa-miR-451a (the first five in serum and the last three in lung tissue). These miRNAs could be used as biomarkers of oral corticosteroid treatment, allowing for the differentiation of patients treated with OCS from the patients not treated with OCS.

## Figures and Tables

**Figure 1 ijms-24-01611-f001:**
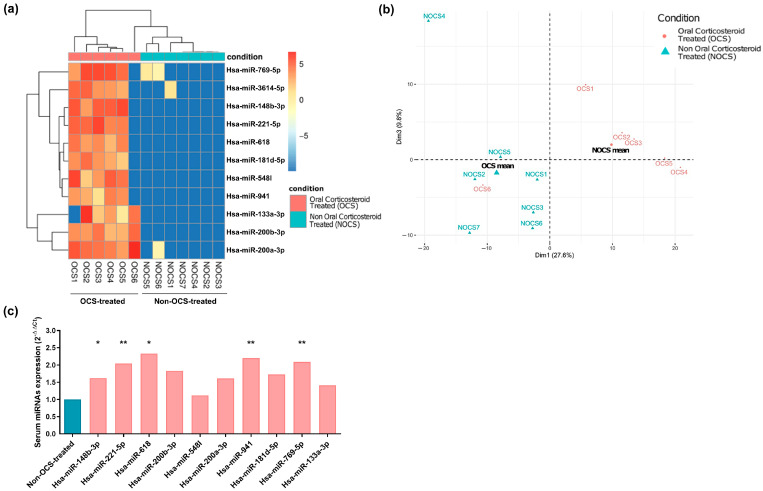
miRNA expression profile in the serum samples from asthmatic patients. (**a**) Heatmap of differentially expressed serum miRNAs between the OCS-treated and non-OCS-treated patients with severe asthma (adjusted *p* < 0.05; Log2 fold change ≥1.5). (**b**) Principal component analysis (PCA) of severe asthmatic serum samples showing the two different treatment groups (OCS-treated vs. non-OCS-treated severe asthmatics). (**c**) RT-qPCR validation of the differentially expressed serum miRNAs between OCS-treated and non-OCS-treated individuals with severe asthma. 2^−∆∆Ct^ values are represented in the graph (**, *p* < 0.01; *, *p* < 0.05).

**Figure 2 ijms-24-01611-f002:**
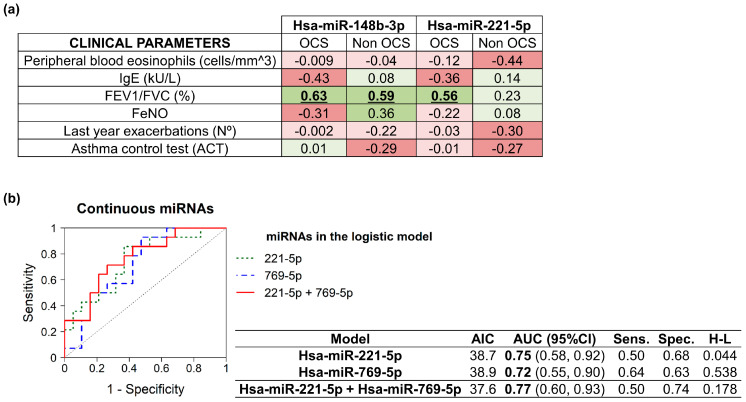
Correlation analyses and the ROC curves of differentially expressed serum miRNAs. (**a**) Table showing the correlation coefficient (r) between the distinct clinical parameters and serum miRNA expression levels. Red color intensity indicates a higher negative correlation, whereas green color intensity shows higher positive correlations. Bold and underlined values were statistically significant (*p* < 0.05). (**b**) ROC curves and multivariate logistic regression models (continuous predictors) of serum hsa-miR-221-5p and hsa-miR-769-5p. Bold values indicate an AUC over 0.70.

**Figure 3 ijms-24-01611-f003:**
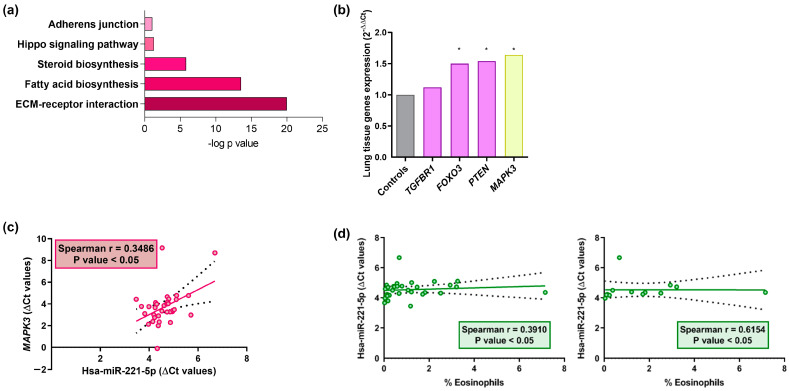
In silico pathway analysis, gene expression, and correlation data of serum miRNA results. (**a**) Graph showing enrichment (as −log *p* value) of the signaling pathways obtained using the DIANA-miRPath v3.0 bioinformatic tool. (**b**) RT-qPCR in the lung biopsies of target genes of the differentially expressed serum miRNAs (hsa-miR-148b-3p and hsa-miR-221-5p). *GAPDH* was used as a housekeeping gene (*, *p* < 0.05). (**c**) The relative miRNA expression of hsa-miR-221-5p (ΔCt) correlated directly with *MAPK3* in the lung tissue samples (*p* < 0.05). (**d**) Correlation graphs of hsa-miR-221-5p in the lung biopsy samples with blood eosinophil percentage (%) in different population groups (asthmatic individuals and total subjects) (*p* < 0.05). Dots (red and green) are individual values, black dashed line indicates error bars and red/green solid line shows linear regression.

**Figure 4 ijms-24-01611-f004:**
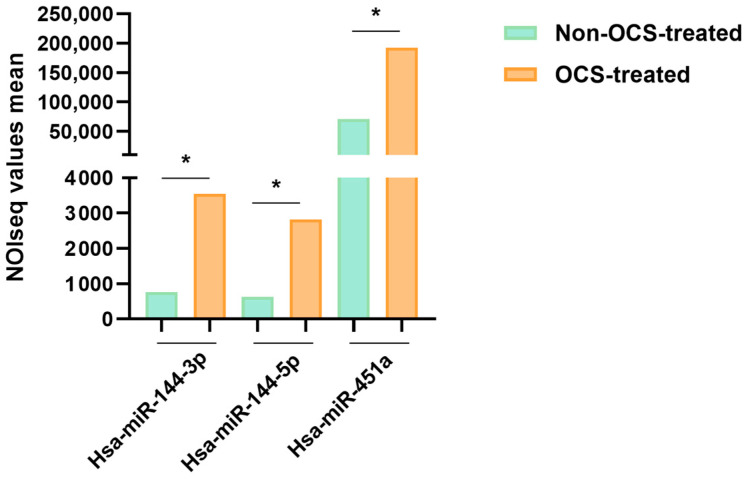
miRNA expression profile in the lung tissue biopsies from asthmatic individuals. Graph showing the mean NOIseq values obtained after miRNAs-Seq bioinformatic analyses (*, *q* ≥ 0.8).

**Table 1 ijms-24-01611-t001:** Demographic, inflammatory, functional, and clinical characteristics of the study subjects (serum sample analysis).

		OCS-Treated (*n* = 20)	Non-OCS-Treated (*n* = 26)	*p*-Value
Age (years) ^†^		54.5 (44.3–59.5)	53.0 (42.8–61.3)	N.S.
Sex (%)	Female	17 (85.0%)	23 (88.5%)	N.S.
BMI ^†^		26.7 (23.1–30.4)	28.7 (24.2–33.4)	N.S.
Smoking habit (%)	Smokers	1 (5.0%)	4/25 (16.0%)	N.S.
	Passive	2 (10.0%)	2/25 (8.0%)	N.S.
	Ex-smokers	7 (35.0%)	9/25 (36.0%)	N.S.
	Non-smokers	10 (50.0%)	10/25 (40.0%)	N.S.
Blood eosinophils (cells/µL) ^†^		300.0 (100.0–500.0)	300.0 (200.0–525.0)	N.S.
Sputum eosinophils (%) ^†^		0.0% (0.0–35.0)	3.1% (2.0–22.1)	N.S.
Atopy (%)		12 (60.0%)	17 (65.4%)	N.S.
IgE (IU) ^†^		366.0 (110.0–690.0)	96.8 (36.6–285.8)	*
FEV_1_/FVC (%) ^#^		67.4% (±15.9)	73.3% (±23.5)	N.S.
FeNO (ppb) ^†^		23.0 (13.5–78.3)	32.0 (13.8–62.0)	N.S.
Exacerbations (%)		17 (85.0%)	16 (61.5%)	N.S.
Severity (%)	Severe	20 (100.0%)	26 (100.0%)	N.S.
	Moderate	0 (0.0%)	0 (0.0%)	N.S.
	Mild	0 (0.0%)	0 (0.0%)	N.S.
	Intermittent	0 (0.0%)	0 (0.0%)	N.S.
ACT ^#^		16.8 (±5.9)	17.2 (±5.5)	N.S.
ICS and LABA (%)		20 (100.0%)	26 (100.0%)	N.S.
Systemic corticosteroid (%)		20 (100.0%)	0 (0.0%)	****

Results are expressed as ^#^ mean (± SD) or ^†^ median (IQR); N.S., non-significant; ****, *p* < 0.0001; *, *p* < 0.05; BMI, body mass index; FEV_1_, forced expiratory volume measured during the first second; FVC, forced vital capacity; FeNO, fractional exhaled nitric oxide; ppb, parts per billion; ACT, asthma control test; ICS and LABA, inhaled corticosteroids and long-acting β2-agonists; OCS, oral corticosteroids.

**Table 2 ijms-24-01611-t002:** Demographic, inflammatory, and clinical characteristics of the study subjects (lung biopsy sample analysis).

		OCS-Treated (*n* = 6)	Non-OCS-Treated (*n* = 7)	*p*-Value
Age (years) ^†^		63.0 (56.8–66.3)	34.0 (27.0–74.0)	N.S.
Sex (%)	Female	3 (50.0%)	1 (14.3%)	N.S.
Smoking habit (%)	Smokers	1 (16.7%)	1/6 (16.7%)	N.S.
	Ex-smokers	3 (50.0%)	4/6 (66.7%)	N.S.
	Non-smokers	2 (33.3%)	1/6 (16.7%)	N.S.
Neutrophils (%) ^†^		61.9% (50.3–72.5)	83.6% (70.6–89.5)	N.S.
Lymphocytes (%) ^†^		24.1% (16.5–35.5)	8.7% (5.9–16.5)	N.S.
Monocytes (%) ^†^		10.0% (8.9–12.2)	5.9% (4.4–9.0)	*
Eosinophils (%) ^†^		2.1% (0.3–3.0)	0.5% (0.1–3.1)	N.S.
Basophils (%) ^†^		0.4% (0.3–0.7)	0.2% (0.1–0.5)	N.S.
Atopy (%)		1 (16.7%)	1 (14.3%)	N.S.
OCS (%)		6 (100.0%)	0 (0.0%)	**

Results are expressed as ^†^ median (IQR); N.S., non-significant; **, *p* < 0.01; *, *p* < 0.05; OCS, oral corticosteroids.

## Data Availability

The data that support the findings of this study are available from the corresponding author, V.d.P., upon reasonable request.

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
