# Peer review of "Analysis of Differentially Expressed MicroRNAs in Serum and Lung Tissues from Individuals with Severe Asthma Treated with Oral Glucocorticoids"

_ijms, 2023, doi:10.3390/ijms24021611_

Round 1

Reviewer 1 Report

In this paper dr Gil-Martinez and coworkers aimed to search for a profile of differentially expressed miRNAs in blood and tissue in severe asthmatics patients receiving OCS treatment versus without OCS therapy using next-generation sequencing (NGS).

General comment:

The data presented is interesting and well described. The authors could add a note in the limitations of the study where they express their interest in possibly evaluating the expression of mirna-regulated proteins with methods such as ELISA (i.e. FOXO3, PTEN, and MAPK3 this aspect would be very interesting, among other things). Beware of minor text editing.

Author Response

Reviewer 1:

In this paper Dr. Gil-Martinez and coworkers aimed to search for a profile of differentially expressed miRNAs in blood and tissue in severe asthmatics patients receiving OCS treatment versus without OCS therapy using next-generation sequencing (NGS).

General comment:

  1. The data presented is interesting and well described. The authors could add a note in the limitations of the study where they express their interest in possibly evaluating the expression of miRNA-regulated proteins with methods such as ELISA (i.e. FOXO3, PTEN, and MAPK3 this aspect would be very interesting, among other things). Beware of minor text editing.

Answer: Dear reviewer, as you have suggested, we have added a note in the limitations section of the study where the interest in evaluating the protein expression levels by ELISA of the genes regulated by 2 of the altered miRNAs (hsa-miR-148b-3p and hsa-miR-221-5p) in this study is stated. This can be seen on page 9, lines 290-292 (document with changes or modifications). In addition, we have corrected minor changes in the text editing.

Reviewer 2 Report

The study by Gil-Martinez et al. describes serum and lung expression levels of a set of miRNAs among subjects with severe asthma treated with oral corticosteroids and those without oral corticosteroids treatment. This is an interesting topic; however, the study has some concerns.

The first one is the number of samples collected, but the authors acknowledged that in the paper. However, because of the low number of samples analyzed, the preliminary nature of the study should be clearly expressed in the manuscript.

The rationale why they selected these specific miRNAs is lacking. For instance, in the “Introduction” section I would focus on the miRNAs most related to the lung inflammation. The authors should cite the work by Mirra et al. (https://doi.org/10.3390/jcm11185446), in which it was shown that the circulating profile of a set of miRNAs, including member of miRNA 181 family, is specifically related to lung inflammation.

Why was GAPDH used as the housekeeping gene and not miRNA 16 or U6 for RT-qPCR validation of differentially expressed serum miRNAs between the groups studied?

In the correlation between serum miR-148 and miR-221 expression levels and clinical features, only the correlation with FEV1/FVC was discussed. Regarding miR-148, it might also be interesting to discuss the correlation with FeNO, which appears to have opposite trends between the two groups.

In-silico analysis should be described in detail in the “Material and Methods” section.

Since miRNAs generally behave as silencers of their target mRNAs, how do you explain the positive correlation between hsa-miR-221-5p and its target gene, MAPK3?

Author Response

Reviewer 2:

The study by Gil-Martinez et al. describes serum and lung expression levels of a set of miRNAs among subjects with severe asthma treated with oral corticosteroids and those without oral corticosteroids treatment. This is an interesting topic; however, the study has some concerns.

  1. The first one is the number of samples collected, but the authors acknowledged that in the paper. However, because of the low number of samples analyzed, the preliminary nature of the study should be clearly expressed in the manuscript.

Answer: As you comment, we have recognised, as a limitation of the study, the low number of lung biopsy samples collected and, as you say, we have added that, due to the limited number of lung biopsy samples analysed, this is a preliminary level study. This can be seen on page 9, lines 290-294 (document with changes or modifications).

  1. The rationale why they selected these specific miRNAs is lacking. For instance, in the “Introduction” section I would focus on the miRNAs most related to the lung inflammation. The authors should cite the work by Mirra et al. (https://doi.org/10.3390/jcm11185446), in which it was shown that the circulating profile of a set of miRNAs, including member of miRNA 181 family, is specifically related to lung inflammation.

Answer: Following this comment, as you suggest, we have cited the work of Mirra et al. (https://doi.org/10.3390/jcm11185446), thus exemplifying the potential of miRNAs as mediators and biomarkers of asthma. This can be seen on page 2, lines 83-86 (document with changes or modifications).

  1. Why was GAPDH used as the housekeeping gene and not miRNA 16 or U6 for RT-qPCR validation of differentially expressed serum miRNAs between the groups studied?

Answer: The endogenous GAPDH gene is used to normalise gene expression, while for RT-qPCR validation of differentially expressed serum miRNAs between the groups studied, hsa-miR-103a-3p, hsa-miR-191-5p, SP6, cel-miR-39-3p, and U6 (Qiagen) were selected as housekeeping miRNAs after we check stability and their good behaviour as housekeeping. The answer to this question can be found in the material and methods section on page 11, lines 375-377 and 390-391 (document with changes or modifications).

  1. In the correlation between serum miR-148 and miR-221 expression levels and clinical features, only the correlation with FEV1/FVC was discussed.Regarding miR-148, it might also be interesting to discuss the correlation with FeNO, which appears to have opposite trends between the two groups.

Answer: As you say, it is interesting to discuss the correlation between hsa-miR-148b-3p expression levels with FeNO. We have therefore added, following your instructions, this information. This can be seen on page 5, lines 152-156 (document with changes or modifications).

  1. In-silico analysis should be described in detail in the “Material and Methods” section.

Answer: According to this comment, we have rewritten in detail the information concerning the In-silico analysis, in the “Material and Methods” section. This can be seen on page 11, lines 393-396 and 397-399 (document with changes or modifications).

  1. Since miRNAs generally behave as silencers of their target mRNAs, how do you explain the positive correlation between hsa-miR-221-5p and its target gene, MAPK3?

Answer: Since, as you say, miRNAs generally behave as silencers of their target mRNAs, an inverse correlation between miRNA-target gene would be expected. However, miRNAs-mRNAs relationships are complex, and the lung tissue samples (in this case) in which they have been studied, miRNA and target gene, differ from each other, and there may be other miRNAs involved in the modulation of a particular mRNA (MAPK3 mRNA, in this case) and also other target genes of this miRNA (hsa-miR-221-5p, in this case) that could be inhibitors of the target gene we are talking about, MAPK3. In this way, we could explain the positive correlation between hsa-miR-221-5p and its target gene, MAPK3, an explanation that is found on page 9, lines 272-281 (document with changes or modifications).

Round 2

Reviewer 2 Report

The authors have addressed all the comments with additional explanations.  I congratulate them for their efforts to implement the comments and suggestions made in the previous version of the manuscript. I belive that the scientific quality of the revised manuscript corresponds to be published in IJMS journal.